# Analyzing Prescribing Trends of Gastric Acid Suppressants in the UK Using Segmented Regression: Evaluating the Influence of COVID-19 (2019–2024)

**DOI:** 10.3390/healthcare13050442

**Published:** 2025-02-20

**Authors:** Khalid Orayj

**Affiliations:** Clinical Pharmacy Department, College of Pharmacy, King Khalid University, Abha 61441, Saudi Arabia; korayg@kku.edu.sa

**Keywords:** gastric acid suppressants, prescribing trends, COVID-19 impact

## Abstract

Background: This study explored the prescribing patterns of proton pump inhibitors (PPIs) and histamine H2-receptor antagonists (H2RAs) across the UK during the COVID-19 pandemic, highlighting the dynamic relationship between emerging evidence, regulatory actions, and clinical practices. Methods: Using a repeated cross-sectional design, prescription data from July 2019 to May 2024 were analyzed across England, Scotland, Wales, and Northern Ireland. Segmented regression analysis was employed to assess trends before and after January 2022, reflecting the impact of emerging evidence on prescribing behaviors. Results: The results revealed a significant increase in famotidine prescriptions, from 57.56 to 303.31 per 100,000 population in England post-January 2022, reflecting early adoption of preliminary findings despite the lack of randomized controlled trial confirmation. Ranitidine prescriptions fell to near zero due to contamination concerns, while PPIs like omeprazole remained the most prescribed, with Wales reporting the highest post-2022 usage at 7445.71 per 100,000 population. Conclusions: Adherence to deprescribing guidelines was inconsistent, with a possibility that many PPI users lacked documented indications. Regional variations in prescribing trends highlighted differences in guideline implementation. These findings underscore the need for improved evidence dissemination and adherence to prescribing guidelines. Future research should include patient-level data and long-term evaluations to optimize healthcare practices.

## 1. Introduction

Proton pump inhibitors (PPIs) and histamine H2-receptor antagonists (H2RAs) are essential therapeutic agents globally, and are primarily used for managing acid-related gastrointestinal disorders such as gastroesophageal reflux disease (GERD), peptic ulcers, and Zollinger–Ellison syndrome [1]. Despite their widespread utility, the global prevalence of their use has raised concerns about their long-term effects. For instance, PPIs rank among the most prescribed medications in countries like the United States and England, with millions of prescriptions issued annually [2]. Similarly, H2RAs like famotidine are frequently employed, not only for their traditional indications but also for their emerging potential in managing diseases like COVID-19 [3].

The intersection of gastric acid suppressant use and COVID-19 has garnered substantial attention due to the unique pathophysiological interactions between these agents and the virus. SARS-CoV-2, the causative agent of COVID-19, utilizes the angiotensin-converting enzyme 2 (ACE2) receptor for cellular entry, which is expressed in both respiratory and gastrointestinal epithelial cells [4]. The virus’s ability to replicate in the gastrointestinal tract and its presence in feces highlight potential gastrointestinal involvement in its pathogenesis [5]. Acid suppression may theoretically increase susceptibility to SARS-CoV-2 by reducing gastric acidity, a natural barrier against pathogens [4].

A growing body of literature, including multiple meta-analyses published in 2021, has explored the implications of PPI and H2RA use in the context of COVID-19 [3,6,7,8,9,10,11]. Notably, PPIs have been associated with increased risks of severe outcomes, such as hospitalization and mortality, possibly due to their role in altering gut microbiota and immune responses [12]. However, few meta-analyses conducted within the same timeframe have refuted this relationship, suggesting that PPIs do not significantly impact the severity of SARS-CoV-2 infection or related outcomes when adjusted for various confounders [5,13]. Despite these latter studies, the medical community remained concerned about this issue, which underscores the need for further investigation into prescribing trends and the impact of these medications during the pandemic. Conversely, H2RAs like famotidine have shown promise in mitigating COVID-19 severity by modulating inflammatory pathways, with some studies reporting decreased risks of intubation and death [8,9].

In the UK, the COVID-19 pandemic catalyzed unprecedented changes in healthcare delivery and prescribing practices, including those for acid-suppressive therapies. While PPIs remained a cornerstone of gastrointestinal management, concerns about their safety in COVID-19 patients may have influenced prescribing trends [14]. Concurrently, interest in the potential protective effects of H2RAs against COVID-19 could have shaped their utilization, highlighting the evolving role of these medications during a global health crisis. These pandemic-driven shifts in prescribing practices occurred against the backdrop of the National Institute for Health and Care Excellence (NICE) guidelines introduced in 2014, which emphasized the minimal effective dose of PPIs and annual medication reviews to curb overuse [15]. However, recent evaluations indicate persistent deviations from these recommendations. A 2024 study found that 62% of patients on continuous PPI therapy lacked a recorded indication for use, and 40% had no medication review within the prior year, reflecting enduring challenges in adhering to deprescribing efforts [15].

The growing body of evidence, including meta-analyses published in 2021 [3,6,7,8,9,10,11], has further underscored the need to reevaluate PPI and H2RA prescribing practices, particularly in the context of the pandemic. These meta-analyses raised concerns regarding the association of PPI use with adverse COVID-19 outcomes while highlighting the potential protective effects of H2Ras.

The current study is designed to determine whether the findings of these meta-analyses have impacted prescribing trends in the UK, specifically examining if PPI prescriptions have decreased or if there has been an increase in H2RA prescriptions. However, it is important to note that the study does not advocate for adherence to meta-analyses; rather, it explores the influence of these analyses on prescribing practices, recognizing that adherence is primarily to clinical guidelines. By analyzing comprehensive prescribing data from all four UK nations, this study aims to provide insights into how recent clinical evidence and pandemic-related factors have shaped real-world prescribing practices. Such an analysis is crucial for optimizing clinical guidelines, ensuring evidence-based prescribing, and improving patient care during and beyond pandemics.

## 2. Materials and Methods

### 2.1. Study Design

This investigation adopted a repeated cross-sectional approach to examine trends in prescribing gastric acid suppressants across the United Kingdom from July 2019 to May 2024. The timeframe was selected to ensure comprehensive data coverage from England, Scotland, Wales, and Northern Ireland. The study concentrated on two primary classes of medications, proton pump inhibitors (PPIs) and H2-receptor antagonists (H2RAs). Table 1 presents the medications included in the analysis along with their *British National Formulary* (BNF) codes. Drugs without any recorded prescriptions during the study period were excluded, with BNF codes used to identify the relevant medications. While some PPIs and H_2_RAs are available over the counter (OTC) at lower doses for short-term relief, higher doses and long-term use require prescriptions. As the study relied solely on GP prescription records, OTC data were unavailable, potentially leading to an underestimation of total usage. However, given that higher doses and prolonged treatments require prescriptions, the data likely capture a majority of sustained clinical use.

Data were sourced from publicly available repositories: OpenPrescribing.net for England [16], NHS Wales Shared Services Partnership for Wales [17], Public Health Scotland’s Monthly Prescribing Activity for Scotland [18], and GP Prescribing Data on Open Data Northern Ireland [19]. These datasets are openly licensed under the Open Government Licence (OGL), and as such, no ethical approval was necessary for this research. The analysis was confined to prescriptions issued by general practitioners in community settings, excluding prescriptions from hospitals or other institutions.

### 2.2. Prevalence Estimation

The prevalence of prescriptions for gastric acid suppressants was calculated by dividing the monthly prescription counts by the corresponding population for each country, then multiplying by 100,000 to standardize rates per 100,000 individuals. Population estimates were derived from official government statistics [20]. However, since population figures for Scotland and Northern Ireland were unavailable for 2023 and 2024 and for all UK nations in 2024, projections based on prior-year growth rates were used for these periods.

### 2.3. Statistical Methods

Interrupted time series (ITS) analysis was employed to evaluate trends in gastric acid suppressant prescriptions over the study period. A linear regression model using the enter method was applied to ensure all variables were included simultaneously in the analysis, allowing a comprehensive assessment of their collective influence. The analysis, conducted using SPSS version 28, accounted for prescribing trends, level changes, and differences across the four UK countries. England was used as the reference category in comparative analyses through interaction terms. The trend examined the effect of time in general, while the level changes classified the time into pre-January 2022 and post-2022.

To address multicollinearity, variance inflation factors (VIFs) were calculated, with values between 5 and 10 deemed acceptable. Variables with VIFs exceeding 10 were excluded, prioritizing the retention of key predictors such as trend and level shifts [21]. Separate models were created for each country, and interaction terms were applied to explore variations in response to findings from 2021 meta-analyses. A significance threshold of *p* < 0.05 was used.

Autocorrelation in the data was managed by incorporating lag variables for each medication. The Durbin–Watson statistic was calculated, with acceptable values between 1.5 and 2.5 [22]. When the statistic indicated serial correlation, higher-order lags were introduced. First-order lags addressed correlations with immediately preceding residuals, while second-order lags accounted for correlations two periods prior.

Additionally, a secondary model assessed delayed effects on prescribing behaviors, incorporating a six-month lag following the publication of the 2021 meta-analyses. This model was instrumental in identifying long-term influences on prescribing trends, classifying the time into pre-July 2022 and post-July 2022.

Through this methodological framework, the study ensured a rigorous evaluation of prescribing patterns by addressing multicollinearity, autocorrelation, and potential delayed impacts. The findings offer insights into regional variations and the broader influence of 2021 meta-analyses on gastric acid suppressant prescriptions in the UK.

## 3. Results

As shown in Table 2, prescribing patterns for gastric acid suppressants varied significantly across regions, with notable differences in the most frequently prescribed drugs before and after January 2022. PPIs were consistently more prescribed than H2RAs across all four UK countries.

Among the PPIs, omeprazole was the most prescribed drug overall. In Wales, it reached the highest level of use, with 7445.71 prescriptions per 100,000 population (95% CI: 7353.93–7537.49) after January 2022. Lansoprazole followed closely and was the second-most prescribed PPI in all countries, with notable increases, such as in Northern Ireland, where it rose from 3250.06 (95% CI: 3187.70–3312.42) to 3537.00 (95% CI: 3485.61–3588.39) (Table 2).

Among H2RAs, famotidine showed the most remarkable growth, particularly in England, where prescriptions surged from 57.56 (95% CI: 36.07–79.05) to 303.31 (95% CI: 271.76–334.85) per 100,000 population. In contrast, ranitidine hydrochloride saw a sharp decline across all regions, dropping to nearly zero after January 2022 (Table 2).

In terms of total prescriptions, Wales had the highest number of PPI prescriptions after January 2022 (13,311.96, 95% CI: 13,153.88–13,470.03), while Scotland saw the most substantial reduction in H2RAs, decreasing from 606.64 (95% CI: 481.83–731.45) to 252.60 (95% CI: 230.17–275.03) (Table 2 and Figure 1 and Figure 2).

Table 2 also reveals regional variations in the overall prescribing trends for both PPIs and H2RAs. For PPIs, Wales consistently reported the highest usage, with total prescriptions rising from 12,587.35 (95% CI: 12,341.61–12,833.10) per 100,000 population before January 2022 to 13,311.96 (95% CI: 13,153.88–13,470.03) after January 2022. In comparison, Scotland showed a notable but smaller increase in total PPI prescriptions from 9240.92 (95% CI: 9030.45–9451.38) to 10,139.33 (95% CI: 9981.98–10,296.69) over the same period. For H2RAs, England exhibited significant changes, with total prescriptions declining from 346.77 (95% CI: 229.12–464.43) before January 2022 to 333.02 (95% CI: 300.93–365.11) thereafter. Similarly, Northern Ireland experienced a decrease in H2RA prescriptions, with a drop from 685.58 (95% CI: 548.10–823.07) to 289.29 (95% CI: 268.99–309.59) post-January 2022.

The results of the segmented regression analyses, presented in Table 3 and Table 4, show significant prescribing trends for gastric acid suppressants between July 2019 and May 2024. These analyses evaluated trends, the impact of the 2021 studies suggesting proton pump inhibitors (PPIs), but not H2RAs, were linked to adverse COVID-19 outcomes, and country-specific differences compared to England. Table 3 focuses on the overall prescribing trends and examines the immediate impact of the 2021 studies, highlighting changes in prescribing practices across the four countries. In contrast, Table 4 shows results of a sensitivity analysis that extends the observation period by six months, accounting for potential lag effects in the adoption of these studies’ findings.

For H2RAs, cimetidine showed a significant declining trend in Scotland (Table 4), with a coefficient of −0.08 (95% CI: −0.151 to −0.01, *p* = 0.025), and in Northern Ireland, where the trend coefficient was −0.101 (95% CI: −0.196 to −0.005, *p* = 0.04). However, the interaction terms comparing these countries to England were not statistically significant, indicating no significant differences in prescribing trends attributable to regional factors relative to England.

Famotidine exhibited a significant positive trend in Wales (0.345 per month, 95% CI: 0.026 to 0.664, *p* = 0.034). Interaction terms revealed significant differences compared to England in Scotland (−5.761, 95% CI: −10.691 to −0.83, *p* = 0.022) and Wales (−5.431, 95% CI: −10.408 to −0.455, *p* = 0.033). Northern Ireland showed a marginally significant interaction term (−4.4, 95% CI: −8.899 to 0.098, *p* = 0.055), suggesting slight differences compared to England.

For ranitidine hydrochloride, trends and interaction terms were not significant across any region. In the combined analysis of all H2RAs, England demonstrated a significant positive trend (1.5 per month, 95% CI: 0.22 to 2.781, *p* = 0.022), as did Scotland (1.45 per month, 95% CI: 0.563 to 2.337, *p* = 0.002). Interaction terms for Scotland, Wales, and Northern Ireland were not significant compared to England.

For PPIs, esomeprazole showed significant prescribing trends in all regions, with Northern Ireland having the highest increase (3.36 per month, 95% CI: 2.42 to 4.3, *p* < 0.001). Compared to England, Wales had a significant negative interaction term (−30.946, 95% CI: −38.122 to −23.77, *p* < 0.001).

Lansoprazole exhibited significant trends across all regions, with England showing the largest increase (28.687 per month, 95% CI: 22.087 to 35.288, *p* < 0.001). Compared to England, Scotland had a significant negative interaction term (−1682.244, 95% CI: −2020.859 to −1343.629, *p* < 0.001), as did Northern Ireland (−1208.672, 95% CI: −1441.802 to −975.542, *p* < 0.001).

Omeprazole trends were significant across all regions, with Northern Ireland showing the largest increase (16.296 per month, 95% CI: 10.959 to 21.633, *p* < 0.001). Interaction terms comparing other countries to England were significant, with Wales showing the most notable difference (2771.469, 95% CI: 2353.285 to 3189.652, *p* < 0.001).

Pantoprazole showed significant increases in all regions, with Northern Ireland exhibiting a strongly positive interaction term (284.247, 95% CI: 226.758 to 341.737, *p* < 0.001) compared to England. Rabeprazole sodium showed significant reductions in interaction terms in all regions compared to England, with Northern Ireland showing the largest difference (−10.08, 95% CI: −12.49 to −7.671, *p* < 0.001).

For all PPIs combined, prescribing trends were significantly positive across all regions, with Northern Ireland showing the highest trend (43.053 per month, 95% CI: 33.545 to 52.561, *p* < 0.001). Interaction terms revealed significant differences compared to England in Scotland (−705.992, 95% CI: −877.544 to −534.441, *p* < 0.001) and Wales (−520.974, 95% CI: −847.483 to −194.466, *p* = 0.002). The inclusion of a six-month sensitivity period did not materially alter these interaction effects, although it enhanced the precision of some estimates.

The adjusted R^2^ values, mostly above 0.8 in most models, demonstrate strong model fit. Durbin–Watson statistics ranged between 1.5 and 2.5, indicating minimal autocorrelation and reliable regression estimates. These metrics confirm the robustness and reliability of the analyses, supporting their relevance for clinical and policy decision-making.

## 4. Discussion

This study examined prescribing trends for PPIs and H2RAs in the UK during the COVID-19 pandemic, providing a detailed assessment of how clinical evidence and pandemic-related challenges shaped community prescribing practices. By incorporating data from all four UK nations, it reveals significant shifts, notably an increase in the use of H2RAs like famotidine and the sustained prevalence of PPIs, despite known risks in COVID-19 patients.

Evidence linking PPIs and H2RAs to COVID-19 outcomes has been extensively explored. PPIs have been associated with higher hospitalization and mortality risks, potentially due to their influence on gut microbiota and immune function [6]. However, the relationship between PPIs and COVID-19 outcomes remains contentious, as other studies have refuted this association, suggesting that PPIs do not significantly impact the severity of SARS-CoV-2 infection or related outcomes when adjusted for various confounders. This ongoing debate underscores the need for further investigation into the impact of PPIs, as the conflicting evidence warrants a deeper examination of their role in the context of COVID-19 [5,6,13]. In contrast, famotidine has been highlighted for its potential anti-inflammatory effects, which may alleviate disease severity [8,9]. However, recent systematic reviews have concluded that famotidine does not significantly reduce mortality or improve recovery rates in COVID-19 patients, suggesting that it should not be used for this indication without confirmation from large-scale randomized controlled trials [23]. Nonetheless, observational studies conducted in 2021 and early 2022 appear to have heavily influenced prescribing behaviors during this period [6,8]. The substantial increase in famotidine use post-January 2022 reflects the early reliance on preliminary evidence, even in the absence of definitive RCT findings.

The trajectory of ranitidine prescribing, by contrast, saw a sharp and immediate decline. This was driven by concerns over its contamination with N-nitrosodimethylamine (NDMA), a potential human carcinogen [24]. Regulatory agencies such as the Food and Drug Administration (FDA) and European Medicines Agency (EMA) identified ranitidine’s inherent instability, which can lead to NDMA accumulation under certain conditions, particularly high temperatures [24]. Unlike other drugs affected by NDMA, where impurities stemmed from manufacturing processes, ranitidine’s unique molecular structure presented a consistent risk, leading to its complete market withdrawal. As a result, prescriptions for ranitidine fell to nearly zero, with alternative therapies such as famotidine and PPIs filling the gap.

In the UK, prescribing practices are guided by authoritative resources such as the *British National Formulary* (BNF), which provides healthcare professionals with comprehensive information on the selection and use of medicines, including indications, contraindications, side effects, doses, and legal classifications [25]. Additionally, NICE offers evidence-based guidelines to support clinical decision-making. NICE recommends that for individuals with gastroesophageal reflux disease (GERD), clinicians should offer a full-dose PPI for 4 or 8 weeks, and if symptoms recur after initial treatment, provide the lowest dose possible to control symptoms [26]. This approach aims to minimize unnecessary medication use and potential side effects, promoting safer prescribing practices.

The pandemic significantly influenced prescribing practices, with shifts in medication utilization reflecting both clinical priorities and external pressures. Omeprazole remained the most prescribed PPI in the UK, a trend reflecting its well-established role in treating gastrointestinal conditions. Regional variations were evident, with Wales reporting the highest rate of PPI prescribing, while Scotland recorded the most substantial decline in H2RA use. These differences likely stem from variations in healthcare delivery, demographic factors, and regional implementation of prescribing guidelines. Notably, the persistence of PPI use in cases lacking regular reviews or documented indications highlights ongoing challenges in achieving adherence to deprescribing recommendations [26].

The study revealed a significant delay in the influence of the 2021 meta-analyses on prescribing patterns, with reductions in proton pump inhibitor (PPI) prescriptions becoming more noticeable after approximately six months. This delay highlights the time required for new evidence to be disseminated and integrated into clinical practice, especially during periods of heightened healthcare demands. While adherence to clinical guidelines is essential, the delay in adapting to the latest evidence may suggest a lack of clinician awareness or access to the most recent meta-analyses and emerging safety concerns. These challenges can be attributed to systemic factors such as organizational inertia, limited dissemination of updated research, and the complexities of aligning new findings with existing clinical workflows [27,28,29]. These barriers underscore the difficulty in translating research findings into actionable changes, which remains a persistent issue across healthcare systems worldwide. To address this difficulty, several approaches to better disseminating new meta-analyses and literature in clinical settings include regular continuing medical education (CME) initiatives, incorporating the latest research into electronic health records (EHRs) for easy access, and fostering better communication across healthcare teams to ensure that new evidence is effectively shared and followed. These measures are crucial for improving adherence to the most up-to-date evidence, ensuring that clinical practice aligns with the latest findings, and ultimately improving patient care [28,29].

Understanding prescribing trends for PPIs and H2RAs in the UK requires considering regional differences in healthcare systems. While the same drugs are available, variations in healthcare delivery, governance, and funding affect prescribing practices. England’s centralized system follows NICE guidelines, ensuring standardized prescribing, while Scotland and Wales, having abolished prescription charges, may have higher prescribing rates due to reduced financial barriers [30]. Scotland’s deprescribing initiatives and reliance on local formularies have led to reduced H2RA use, while Northern Ireland’s integrated health and social care system introduces unique prescribing dynamics [31,32]. Despite the increase in PPI prescribing, rates have plateaued in recent years due to awareness of long-term risks, whereas H2RAs, with a safer profile for long-term use, are seeing a modest resurgence [14]. These regional differences underscore the need for tailored interventions to optimize prescribing practices based on local healthcare structures.

This study has several notable strengths. It is the first investigation to evaluate prescribing trends for gastric acid suppressants across all four UK nations, offering a uniquely comprehensive view of community-level prescribing during the pandemic. The reliance on GP prescribing data provides a robust representation of real-world medication use, which is critical for understanding primary care practices. While much of the existing literature on gastric acid suppressants in the context of COVID-19 is based on hospital data [6,12], this study focuses on community prescribing, recognizing the potential spillover effects of hospital-based findings on general practice.

However, the study also has limitations. In addition to excluding hospital prescribing data, which may underestimate the total use of these medications, particularly among severely ill COVID-19 patients, the study did not have access to over-the-counter (OTC) medication data. Furthermore, the study was unable to obtain information regarding the ratio of hospital prescribing to community prescribing, which limits the overall understanding of medication usage across different settings. This lack of data further restricts the comprehensiveness of the study’s findings. The lack of patient-level data, such as demographics, comorbidities, and COVID-19 status, limits the ability to investigate specific drivers of prescribing changes. Additionally, the reliance on population estimates for the most recent years (2023–2024) introduces some uncertainty to prevalence calculations, though overall trends remain consistent. The single-author nature of this study also limits the inclusion of multidisciplinary insights from specialists in gastroenterology, internal medicine, or UK healthcare systems. While the author has experience analyzing UK prescribing data, a team-based approach would enhance future research by integrating diverse expertise. Addressing these limitations in future studies, along with extending the scope to assess long-term implications of prescribing patterns, would provide a more comprehensive understanding of their effects on patient outcomes and healthcare systems.

In conclusion, this study highlights the need for greater adherence to clinical guidelines, including deprescribing efforts for PPIs and appropriate use of H2RAs. The rapid decline in ranitidine prescribing reflects the effectiveness of regulatory actions in response to safety concerns, while the rise in famotidine use demonstrates how early observational studies can shape prescribing behaviors even in the absence of definitive evidence. Future research should address the study’s limitations by including hospital data, patient-level insights, and long-term evaluations to provide a more complete understanding of the consequences of these trends.

## Figures and Tables

**Figure 1 healthcare-13-00442-f001:**
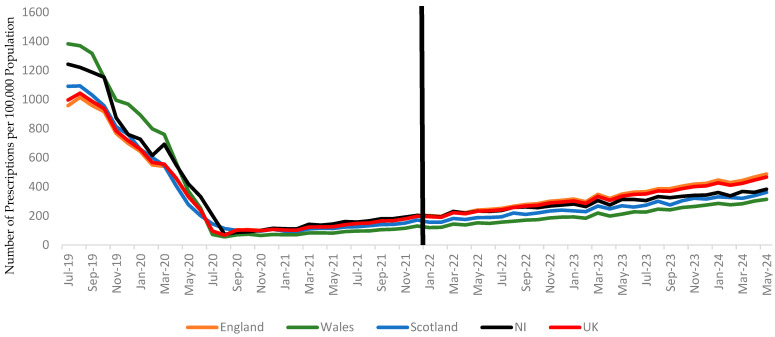
Trends in histamine 2-receptor antagonist (H2RA) prescriptions per 100,000 population in the UK, with a vertical line distinguishing the periods before and after January 2022. The data represent prescription trends in England, Wales, Scotland, and Northern Ireland.

**Figure 2 healthcare-13-00442-f002:**
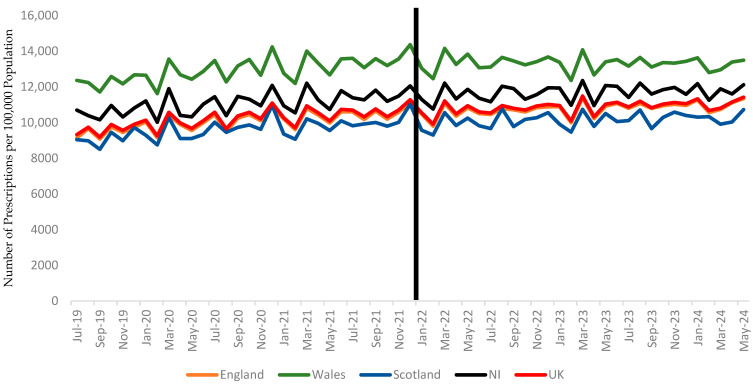
Trends in proton pump inhibitor (PPI) prescriptions per 100,000 population in the UK, with a vertical line distinguishing the periods before and after January 2022. The data represent prescription trends in England, Wales, Scotland, and Northern Ireland.

**Table 1 healthcare-13-00442-t001:** Gastric acid suppressants and corresponding BNF codes.

Gastric Acid Suppressant Category	Medications and BNF Codes
Histamine 2-receptor antagonists (H2RAs)	Cimetidine: 0103010D0
	Famotidine: 0103010H0
	Nizatidine: 0103010N0
	Ranitidine hydrochloride: 0103010T0
Proton pump inhibitors (PPIs)	Esomeprazole: 0103050E0
	Lansoprazole: 0103050L0
	Omeprazole: 0103050P0
	Pantoprazole: 0103050R0
	Rabeprazole sodium: 0103050T0

**Table 2 healthcare-13-00442-t002:** Comparison of monthly average gastric acid suppressants prescriptions per 100,000 population across England, Scotland, Wales, and Northern Ireland before and after January 2022, with 95% confidence intervals.

	England		Scotland		Wales		Northern Ireland	
Gastric Acid Suppressants	Before January 2022 *	After January 2022 **	Before January 2022 *	After January 2022 **	Before January 2022 *	After January 2022 **	Before January 2022 *	After January 2022 **
Cimetidine	14.32 (10.47–18.17)	10.67 (10.42–10.92)	12 (9.48–14.52)	10.01 (9.8–10.23)	11.51 (8.97–14.05)	12.43 (12.05–12.82)	17.52 (13.68–21.37)	13.73 (13.31–14.15)
Famotidine	57.56 (36.07–79.05)	303.31 (271.76–334.85)	31.53 (18.45–44.62)	231.8 (209.38–254.23)	21.09 (12.75–29.42)	164.37 (145.04–183.7)	38.72 (22.99–54.44)	250.73 (231.13–270.32)
Nizatidine	10.54 (7.99–13.09)	18.76 (18.28–19.23)	5.07 (3.43–6.7)	10.75 (10.56–10.94)	9.41 (6.98–11.83)	31.89 (29.92–33.85)	10.65 (8.24–13.07)	24.8 (24.14–25.46)
Ranitidine hydrochloride	264.35 (130.9–397.8)	0.29 (0.18–0.4)	558.05 (421.09–695)	0.03 (0.01–0.05)	657.91 (480.17–835.65)	0.14 (0.05–0.23)	618.69 (466.16–771.22)	0.04 (0.02–0.05)
All H2RAs	346.77 (229.12–464.43)	333.02 (300.93–365.11)	606.64 (481.83–731.45)	252.6 (230.17–275.03)	699.91 (531.11–868.71)	208.83 (187.29–230.37)	685.58 (548.1–823.07)	289.29 (268.99–309.59)
Esomeprazole	344.34 (335.19–353.49)	388.67 (382.68–394.67)	492.77 (477.16–508.38)	600.34 (588.25–612.42)	293.45 (282.83–304.07)	367.98 (362.01–373.95)	1072.85 (1046.45–1099.25)	1207.21 (1189.87–1224.55)
Lansoprazole	4387.77 (4289.66–4485.88)	4912.23 (4826.91–4997.54)	2648.59 (2586.39–2710.8)	2955.24 (2903.29–3007.2)	4860.36 (4755.83–4964.88)	5272.8 (5207.43–5338.17)	3250.06 (3187.7–3312.42)	3537 (3485.61–3588.39)
Omeprazole	5069.58 (4981.03–5158.14)	5173.26 (5113.86–5232.65)	5974.65 (5843.76–6105.54)	6454.93 (6358.77–6551.09)	7217.54 (7089.36–7345.72)	7445.71 (7353.93–7537.49)	5805.43 (5685.59–5925.26)	6367.17 (6276.92–6457.41)
Pantoprazole	221.59 (215.47–227.72)	255.79 (251.19–260.39)	96.56 (94.47–98.66)	105.16 (103.41–106.9)	184.74 (181.02–188.46)	200.8 (198.12–203.48)	543.72 (536.19–551.26)	537.82 (530–545.64)
Rabeprazole sodium	54.65 (53.81–53.81)	41.31 (38.93–38.93)	28.34 (27.89–27.89)	23.67 (22.77–22.77)	31.27 (30.79–30.79)	24.67 (23.69–23.69)	40.09 (39.39–39.39)	29.81 (28.35–28.35)
All PPIs	10,077.94 (9877.44–10,278.45)	10,771.26 (10,629.54–10,912.98)	9240.92 (9030.45–9451.38)	10,139.33 (9981.98–10,296.69)	12,587.35 (12,341.61–12,833.1)	13,311.96 (13,153.88–13,470.03)	10,712.15 (10,497.87–10,926.43)	11,678.99 (11,515.31–11,842.68)

* Monthly average number of prescriptions per 100,000 population between July 2019 and January 2022 (31 months), including 95% confidence interval (CI). ** Monthly average number of prescriptions per 100,000 population between January 2022 and May 2024 (29 months), including 95% confidence interval (CI).

**Table 3 healthcare-13-00442-t003:** Segmented regression analysis of gastric acid suppressant trends examining the impact of 2021 studies suggesting PPIs, but not H2RAs, were linked to increased adverse COVID-19 outcomes.

Gastric Acid Suppressants Are Listed in the Rows, While the Prescribing Factors Are Detailed in the Columns	Country	Complete Trend Analysis (July 2019–May 2024) *	The Impact of 2021 Studies Suggesting PPIs, but Not H2RAs, Were Linked to Increased Adverse COVID-19 Outcomes (Binary Variable Distinguishing Pre-January 2022 and Post-January 2022)	Effect of Country Compared to the Reference (England), Following the Interaction Term That Multiplied the Impact of 2021 Studies with the Country	Adjusted R^2^	Durbin–Watson Statistics
Cimetidine	England	−0.075 (−0.19–0.04) (0.195)	1.514 (−2.278–5.306) (0.427)	NA	0.874	1.61 (after first-order lag)
	Scotland	−0.07 (−0.143–0.003) (0.062)	1.358 (−1.07–3.785) (0.27)	−0.085 (−1.299–1.129) (0.889)	0.803	1.75 (after first-order lag)
	Wales	−0.059 (−0.138–0.02) (0.142)	1.25 (−1.392–3.892) (0.351)	0.079 (−1.242–1.401) (0.906)	0.767	1.77 (after second-order lag)
	NI **	−0.092 (−0.191–0.008) (0.07)	1.908 (−1.393–5.209) (0.255)	0.557 (−1.145–2.26) (0.518)	0.804	1.61 (after first-order lag)
Famotidine	England	Removed (VIF = 46.27)	−3.072 (−14.156–8.012) (0.581)	NA	0.995	2.48 (after first-order lag)
	Scotland	Removed (VIF = 24.54)	−5.142 (−13.03–2.746) (0.199)	−5.49 (−10.339–−0.64) (0.027)	0.993	2.25 (after first-order lag)
	Wales	0.424 (0.09–0.759) (0.013)	−4.497 (−11.147–2.154) (0.183)	−5.358 (−10.248–−0.468) (0.032)	0.994	2.45 (after third-order lag)
	NI	1.01 (0.369–1.651) (0.002)	−4.299 (−12.524–3.926) (0.303)	−4.283 (−8.755–0.19) (0.06)	0.992	2.31 (after first-order lag)
Nizatidine	England	0.073 (−0.018–0.164) (0.114)	−0.923 (−3.394–1.549) (0.457)	NA	0.859	1.72 (after first-order lag)
	Scotland	0.046 (−0.01–0.101) (0.105)	−0.8 (−2.43–0.829) (0.332)	−1.079 (−2.052–−0.106) (0.03)	0.869	1.97 (after first-order lag)
	Wales	0.092 (0.011–0.172) (0.026)	−0.718 (−2.785–1.348) (0.493)	1.539 (0.251–2.827) (0.02)	0.922	1.51 (after first-order lag)
	NI	0.077 (0.008–0.145) (0.028)	−1.123 (−2.946–0.699) (0.225)	0.846 (−0.166–1.859) (0.101)	0.899	1.55 (after first-order lag)
Ranitidine hydrochloride	England	0.46 (−0.942–1.863) (0.513)	3.283 (−35.453–42.019) (0.866)	NA	0.983	2.21 (after first-order lag)
	Scotland	0.526 (−0.415–1.467) (0.27)	3.345 (−22.677–29.367) (0.799)	0.497 (−11.757–12.75) (0.936)	0.986	1.64 (after first-order lag)
	Wales	0.551 (−0.671–1.772) (0.374)	4.896 (−29.192–38.984) (0.776)	−1.498 (−17.679–14.684) (0.855)	0.983	1.81 (after first-order lag)
	NI	0.431 (−0.812–1.675) (0.493)	5.774 (−28.745–40.292) (0.741)	−0.173 (−16.47–16.125) (0.983)	0.979	1.72 (after first-order lag)
All H2Ras **	England	1.247 (0.008–2.486) (0.049)	−1.943 (−42.178–38.291) (0.923)	NA	0.972	1.69 (after first-order lag)
	Scotland	1.175 (0.322–2.027) (0.007)	−2.678 (−29.817–24.461) (0.845)	−3.711 (−16.534–9.112) (0.568)	0.978	1.52 (after first-order lag)
	Wales	1.182 (0.097–2.266) (0.033)	−0.554 (−34.522–33.414) (0.974)	−7.896 (−23.988–8.196) (0.333)	0.977	1.62 (after first-order lag)
	NI	1.054 (−0.066–2.174) (0.065)	0.954 (−34.55–36.458) (0.958)	−2.836 (−19.569–13.896) (0.738)	0.967	1.60 (after first-order lag)
Esomeprazole	England	2.115 (1.524–2.705) (<0.001)	−7.851 (−21.913–6.212) (0.268)	NA	0.793	2.04 (after first-order lag)
	Scotland	2.502 (1.866–3.137) (<0.001)	−12.302 (−28.881–4.277) (0.144)	Removed (VIF−19.74)	0.955	1.99 (after first-order lag)
	Wales	2.062 (1.615–2.508) (<0.001)	−5.541 (−16.114–5.033) (0.301)	−29.614 (−36.847–−22.381) (<0.001)	0.837	2.02 (after second-order lag)
	NI	2.944 (2.008–3.881) (<0.001)	−8.939 (−37.519–19.64) (0.537)	957.653 (815.846–1099.46) (<0.001)	0.991	2.01 (after first-order lag)
Lansoprazole	England	30.08 (23.589–36.571) (<0.001)	−138.923 (−286.329–8.483) (0.064)	NA	0.840	2.35 (after first-order lag)
	Scotland	15.124 (10.638–19.61) (<0.001)	−115.815 (−239.553–7.923) (0.066)	−1690.711 (−2023.486–−1357.936) (<0.001)	0.968	2.01 (after first-order lag)
	Wales	19.499 (14.354–24.645) (<0.001)	−153.874 (−296.683–−11.064) (0.035)	546.81 (431.184–662.437) (<0.001)	0.785	1.99 (after second-order lag)
	NI **	15.177 (10.552–19.802) (<0.001)	−98.661 (−228.585–31.263) (0.135)	−1210.882 (−1441.366–−980.398) (<0.001)	0.930	1.97 (after first-order lag)
Omeprazole	England	9.381 (3.45–15.313) (0.003)	−153.649 (−345.382–38.084) (0.114)	NA	0.835	2.04 (after first-order lag)
	Scotland	13.828 (8.303–19.354) (<0.001)	−203.494 (−381.738–−25.249) (0.026)	1408.294 (1178.201–1638.388) (<0.001)	0.867	2.06 (after first-order lag)
	Wales	9.934 (4.336–15.533) (<0.001)	−208.091 (−392.928–−23.255) (0.028)	2573.318 (2160.222–2986.415) (<0.001)	0.955	2.05 (after second-order lag)
	NI	14.601 (9.178–20.023) (<0.001)	−154.552 (−325.503–16.399) (0.076)	1272.33 (1067.982–1476.678) (<0.001)	0.850	2.01 (after first-order lag)
Pantoprazole	England	1.589 (1.172–2.006) (<0.001)	−4.964 (−13.942–4.013) (0.273)	NA	0.848	2.05 (after first-order lag)
	Scotland	0.45 (0.213–0.688) (<0.001)	−3.829 (−10.784–3.126) (0.278)	−70.984 (−93.07–−48.898) (<0.001)	0.982	2.41 (after first-order lag)
	Wales	0.689 (0.42–0.959) (<0.001)	−4.622 (−12.178–2.934) (0.228)	−31.809 (−40.304–−23.313) (<0.001)	0.864	2.15 (after first-order lag)
	NI	0.58 (0.101–1.06) (0.018)	−6.93 (−22.774–8.913) (0.388)	283.003 (225.917–340.089) (<0.001)	0.980	1.99 (after first-order lag)
Rabeprazole sodium	England	−0.023 (−0.124–0.078) (0.647)	−3.87 (−7.002–−0.738) (0.016)	NA	0.877	2.45 (after first-order lag)
	Scotland	−0.011 (−0.07–0.048) (0.707)	−2.451 (−4.353–−0.55) (0.012)	−5.499 (−8.179–−2.819) (<0.001)	0.959	2.36 (after first-order lag)
	Wales	−0.014 (−0.071–0.043) (0.63)	−2.626 (−4.46–−0.791) (0.005)	−5.268 (−7.756–−2.781) (<0.001)	0.958	2.47 (after first-order lag)
	NI	−0.049 (−0.117–0.019) (0.156)	−3.302 (−5.392–−1.213) (0.002)	−5.002 (−7.041–−2.962) (<0.001)	0.919	2.42 (after first-order lag)
All PPIs **	England	−3.128 (−10.507–4.251) (0.399)	243.815 (37.586–450.043) (0.021)	NA	0.811	1.75 (after first-order lag)
	Scotland	38.25 (28.697–47.803) (<0.001)	−362.31 (−646.339–−78.281) (0.013)	−689.574 (−859.95–−519.197) (<0.001)	0.831	2.11 (after first-order lag)
	Wales	33.655 (22.784–44.526) (<0.001)	−385.11 (−719.332–−50.888) (0.024)	3124.624 (2619.234–3630.014) (<0.001)	0.904	2.04 (after first-order lag)
	NI	40.828 (31.255–50.4) (<0.001)	−305.261 (−589.092–−21.431) (0.035)	1302.676 (1090.791–1514.56) (<0.001)	0.832	2.22 (after first-order lag)

* Coefficients from regression analysis are presented alongside their 95% confidence intervals and *p* values. ** NI stands for Northern Ireland, H2RAs for histamine 2-receptor antagonists, PPIs for proton pump inhibitors, and VIF for variance inflation factor.

**Table 4 healthcare-13-00442-t004:** Segmented regression analysis of gastric acid suppressant trends examining the impact of 2021 studies suggesting PPIs, but not H2RAs, were linked to increased adverse COVID-19 outcomes: a sensitivity analysis including 6 months’ post-2021 studies.

Gastric Acid Suppressants Are Listed in the Rows, While the Prescribing Factors Are Detailed in the Columns.	Country	Complete Trend Analysis (July 2019–May 2024) *	The Impact of 2021 Studies Suggesting PPIs, but Not H2RAs, Were Linked to Increased Adverse COVID-19 Outcomes (Binary Variable Distinguishing Pre-January 2022 and Post-January 2022)	Effect of Country Compared to the Reference (England), Following the Interaction Term That Multiplies the Impact of 2021 Studies with the Country	Adjusted R^2^	Durbin–Watson Statistics
Cimetidine	England	−0.086 (−0.197–0.025) (0.126)	1.958 (−1.715–5.63) (0.29)	NA	0.854	1.85 (after first-order lag)
	Scotland	−0.08 (−0.151–−0.01) (0.025)	1.803 (−0.539–4.146) (0.13)	−0.091 (−1.299–1.117) (0.881)	0.805	1.73 (after first-order lag)
	Wales	−0.068 (−0.143–0.007) (0.076)	1.645 (−0.903–4.193) (0.204)	0.078 (−1.239–1.395) (0.906)	0.765	1.77 (after second-order lag)
	NI **	−0.101 (−0.196–−0.005) (0.04)	2.294 (−0.899–5.487) (0.157)	0.594 (−1.103–2.291) (0.489)	0.805	1.63 (after first-order lag)
Famotidine	England	Removed (VIF = 47.43)	2.933 (−8.458–14.323) (0.608)	NA	0.995	2.46 (after first-order lag)
	Scotland	Removed (VIF = 23.58)	−0.251 (−8.063–7.561) (0.949)	−5.761 (−10.691–−0.83) (0.022)	0.993	2.24 (after first-order lag)
	Wales	0.345 (0.026–0.664) (0.034)	−1.281 (−7.825–5.263) (0.699)	−5.431 (−10.408–−0.455) (0.033)	0.993	2.44 (after third-order lag)
	NI	0.966 (0.328–1.605) (0.003)	−1.265 (−9.252–6.722) (0.754)	−4.4 (−8.899–0.098) (0.055)	0.992	2.30 (after first-order lag)
Nizatidine	England	0.07 (−0.028–0.168) (0.157)	−0.732 (−3.222–1.757) (0.558)	NA	0.859	1.70 (after first-order lag)
	Scotland	0.04 (−0.017–0.097) (0.166)	−0.572 (−2.2–1.056) (0.488)	−1.106 (−2.092–−0.12) (0.028)	0.868	1.97 (after first-order lag)
	Wales	0.084 (0.001–0.167) (0.048)	−0.368 (−2.381–1.645) (0.718)	1.556 (0.259–2.853) (0.019)	0.922	1.50 (after first-order lag)
	NI	0.076 (0.002–0.151) (0.045)	−0.98 (−2.841–0.881) (0.299)	0.902 (−0.127–1.931) (0.085)	0.899	1.55 (after first-order lag)
Ranitidine hydrochloride	England	0.898 (−0.619–2.416) (0.241)	−10.469 (−51.297–30.358) (0.609)	NA	0.983	2.22 (after first-order lag)
	Scotland	1.012 (−0.004–2.028) (0.051)	−11.905 (−39.296–15.485) (0.391)	0.43 (−11.788–12.648) (0.945)	0.986	1.63 (after first-order lag)
	Wales	1.079 (−0.23–2.388) (0.105)	−11.975 (−47.672–23.723) (0.508)	−1.85 (−18.021–14.32) (0.821)	0.983	1.81 (after first-order lag)
	NI	0.905 (−0.436–2.246) (0.184)	−9.29 (−45.613–27.033) (0.613)	−0.356 (−16.648–15.937) (0.966)	0.979	1.67 (after first-order lag)
All H2Ras **	England	1.5 (0.22–2.781) (0.022)	−11.407 (−53.878–31.065) (0.592)	NA	0.972	1.71 (after first-order lag)
	Scotland	1.45 (0.563–2.337) (0.002)	−12.656 (−41.211–15.899) (0.382)	−3.493 (−16.282–9.296) (0.589)	0.978	1.51 (after first-order lag)
	Wales	1.541 (0.413–2.669) (0.008)	−13.327 (−48.834–22.181) (0.459)	−7.744 (−23.8–8.311) (0.341)	0.977	1.60 (after first-order lag)
	NI	1.371 (0.208–2.534) (0.021)	−10.483 (−47.734–26.767) (0.578)	−2.912 (−19.624–13.799) (0.731)	0.968	1.62 (after first-order lag)
Esomeprazole	England	2.276 (1.661–2.891) (<0.001)	−11.977 (−25.564–1.61) (0.083)	NA	0.803	2.06 (after first-order lag)
	Scotland	2.47 (1.817–3.123) (<0.001)	−10.207 (−26.33–5.915) (0.212)	Removed (VIF−19.87)	0.955	1.99 (after first-order lag)
	Wales	2.304 (1.825–2.783) (<0.001)	−12.023 (−22.246–−1.8) (0.022)	−30.946 (−38.122–−23.77) (<0.001)	0.838	2.01 (after second-order lag)
	NI	3.36 (2.42–4.3) (<0.001)	−24.015 (−51.449–3.419) (0.086)	977.351 (835.546–1119.155) (<0.001)	0.991	2.01 (after first-order lag)
Lansoprazole	England	28.687 (22.087–35.288) (<0.001)	−87.192 (−232.594–58.21) (0.235)	NA	0.834	2.29 (after first-order lag)
	Scotland	12.992 (8.551–17.434) (<0.001)	−36.865 (−157.768–84.039) (0.547)	−1682.244 (−2020.859–−1343.629) (<0.001)	0.967	2.01 (after first-order lag)
	Wales	19.264 (14.008–24.52) (<0.001)	−137.922 (−278.307–2.464) (0.054)	558.445 (440.853–676.037) (<0.001)	0.775	1.98 (after second-order lag)
	NI **	13.659 (9.116–18.202) (<0.001)	−42.38 (−168.598–83.837) (0.507)	−1208.672 (−1441.802–−975.542) (<0.001)	0.929	1.97 (after first-order lag)
Omeprazole	England	11.089 (5.313–16.866) (<0.001)	−217.453 (−403.251–−31.655) (0.023)	NA	0.842	2.07 (after first-order lag)
	Scotland	14.73 (9.263–20.196) (<0.001)	−235.451 (−409.752–−61.151) (0.009)	1463.218 (1229.599–1696.837) (<0.001)	0.869	2.06 (after first-order lag)
	Wales	13.12 (7.684–18.555) (<0.001)	−330.789 (−510.726–−150.851) (<0.001)	2771.469 (2353.285–3189.652) (<0.001)	0.958	2.08 (after second-order lag)
	NI	16.296 (10.959–21.633) (<0.001)	−215.501 (−381.057–−49.944) (0.011)	1315.718 (1110.258–1521.178) (<0.001)	0.855	2.01 (after first-order lag)
Pantoprazole	England	1.508 (1.088–1.928) (<0.001)	−1.901 (−10.647–6.845) (0.665)	NA	0.845	2.04 (after first-order lag)
	Scotland	0.354 (0.124–0.585) (0.003)	−0.156 (−6.878–6.566) (0.963)	−70.666 (−92.899–−48.434) (<0.001)	0.982	2.42 (after first-order lag)
	Wales	0.656 (0.381–0.93) (<0.001)	−3.135 (−10.506–4.236) (0.401)	−32.233 (−40.84–−23.626) (<0.001)	0.863	2.15 (after first-order lag)
	NI	0.521 (0.062–0.981) (0.027)	−4.763 (−20.08–10.554) (0.539)	284.247 (226.758–341.737) (<0.001)	0.980	1.99 (after first-order lag)
Rabeprazole sodium	England	−0.008 (−0.073–0.057) (0.816)	−13.791 (−17.708–−9.874) (<0.001)	NA	0.929	2.43 (after first-order lag)
	Scotland	−0.006 (−0.056–0.045) (0.823)	−4.163 (−6.34–−1.986) (<0.001)	−8.349 (−11.474–−5.225) (<0.001)	0.962	2.44 (after first-order lag)
	Wales	−0.01 (−0.058–0.037) (0.67)	−4.983 (−7.194–−2.773) (<0.001)	−8.949 (−11.979–−5.919) (<0.001)	0.962	2.35 (after first-order lag)
	NI	−0.047 (−0.097–0.004) (0.068)	−8.802 (−11.351–−6.254) (<.001)	−10.08 (−12.49–−7.671) (<0.001)	0.938	2.35 (after first-order lag)
All PPIs **	England	43.393 (30.385–56.401) (<0.001)	−350.674 (−700.749–−0.6) (0.05)	NA	0.809	2.18 (after first-order lag)
	Scotland	38.139 (28.484–47.793) (<0.001)	−345.605 (−624.113–−67.097) (0.015)	−705.992 (−877.544–−534.441) (<0.001)	0.829	2.14 (after first-order lag)
	Wales	37.916 (26.951–48.882) (<0.001)	−520.974 (−847.483–−194.466) (0.002)	3297.163 (2784.521–3809.804) (<0.001)	0.908	2.06 (after first-order lag)
	NI	43.053 (33.545–52.561) (<0.001)	−376.762 (−650.798–−102.725) (0.007)	1335.249 (1123.669–1546.83) (<0.001)	0.831	2.24 (after first-order lag)

* Coefficients from regression analysis are presented alongside their 95% confidence intervals and *p* values. ** NI stands for Northern Ireland, H2RAs for histamine 2-receptor antagonists, PPIs for proton pump inhibitors, and VIF for variance inflation factor.

## Data Availability

All data used in this study are publicly available and can be accessed by anyone. The datasets analyzed were sourced from open repositories, including OpenPrescribing.net for England (Available at: https://openprescribing.net/ (accessed on 27 September, 2024)), NHS Wales Shared Services Partnership for Wales, Public Health Scotland’s Monthly Prescribing Activity, and GP Prescribing Data on Open Data Northern Ireland. These resources are openly licensed under the Open Government Licence (OGL) and can be accessed without restriction. No new data were generated for this study.

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
