# Peer review of "Analyzing Prescribing Trends of Gastric Acid Suppressants in the UK Using Segmented Regression: Evaluating the Influence of COVID-19 (2019–2024)"

_healthcare, 2025, doi:10.3390/healthcare13050442_

Round 1

Reviewer 1 Report

Comments and Suggestions for Authors

A brief summary

The manuscript analyses the use of PPIs and H2receptor blocking drugs prescribed in the UK.

The major weakness of the manuscript is that the author does not have a co-author who is an expert in UK healthcare, medication use, or a specialist in gastroenterology in the field. Medication utilisation research is always a team work, it is important that the analytical team includes someone who has a complete background knowledge of the data source and is familiar with the health care and prescribing practices of the country. In addition, the analytical team should include a representative of the medical speciality of the group, in this case an internal medicine or gastroenterology specialist.

General concept comments
Article:

The results section in the manuscript does not adequately explain the results of each model, and it would be important to make this clear.

Specific comments

Minor comments:

In the title of Table 2, I think the word "monthly" is omitted, i.e. it is a monthly mean. It would be useful to state how many months it is an average of.

It is necessary to explain these results in detail in the Result section.

Table 4. Unfortunately not understandable, many need more explanation.

Figure 1 .

H2RBs are not visible in the figure, the colouring is confusing.

This figure should be split into 2 figures. The first would have the PPI and the second would have the H2RA blockers. With different scales of Y axis.

Are the exact same drugs available in each region (NI, England, Wales, Scotland)? What are the differences and similarities in the healthcare system in each province? A summary table would be useful.

What about over-the-counter (OTC) PPIs and H2RBs?

Based on the above, I suggest revising this valuable manuscript with the involvement of additional co-authors who are relevant to the analysis of UK medicine use.

  • Is the manuscript clear, relevant for the field and presented in a well-structured manner? 

Partly.

  • Are the cited references mostly recent publications (within the last 5 years) and relevant? Does it include an excessive number of self-citations?

I checked and there is no  self-citations, and cited references mostly recent publications.

  • Is the manuscript scientifically sound and is the experimental design appropriate to test the hypothesis?

Partly

  • Are the manuscript’s results reproducible based on the details given in the methods section?

Yes

  • Are the figures/tables/images/schemes appropriate? Do they properly show the data? Are they easy to interpret and understand? Is the data interpreted appropriately and consistently throughout the manuscript? Please include details regarding the statistical analysis or data acquired from specific databases.

Partly

  • Are the conclusions consistent with the evidence and arguments presented?

·         Partly

  • Please evaluate the ethics statements and data availability statements to ensure they are adequate.

Appropriate

Reviewer 2 Report

Comments and Suggestions for Authors

Dear colleague!

I read with interest your manuscript "Analyzing Prescribing Trends of Gastric Acid Suppressants in the UK Using Segmented Regression: Evaluating the Influence of COVID-19 (2019–2024)" submitted to the Healthcare. It is based on the cross-sectional study evaluating prescribing trends for PPIs and H2RAs in the UK with special emphasis on the influence COVID-19 outbreak and recent data supportive for PPIs overutilization. The data included to the analysis were obtained in primary care, the largest segment of antisecretory drugs prescription. Evidently, the paper brings new to the field and may be interesting to the readers.

However, I would recommend some updates to the manuscript so it was more consistent. There is really a problem of PPIs overutilization with no certain indications for the administration or long-term use. Thats why, current analysis and discussion are necessary. However, despite first meta-analyses showed direct relationship between PPIs use and risks of infection, being intubated and worse outcomes of COVID-19 infection, updated and larger works (like one published in Clinical Gastroenterology and Hepatology in 2021 and in Jpn J Infect Dis 2022 that taken into the account different confounders) concluded that PPIs themselves do not have a significant impact on SARS-CoV-2 infection and related severe outcomes. In this regard, some updates should be made to the introduction and the discussion. 

Related to lines 126-133, most of the physicians follow clinical guidelines, not meta-analyses. Please, check whether some clinical guidelines updates on the PPIs use were issued on the national level at this time. The effect on PPI use may be related to this fact. 

Nevertheless, these comments are more related to the interpretation of the data. This in no way diminishes the dignity of the work itself and the evident need to discuss an appropriate use of any drugs, even relatively safe ones like PPIs.

Reviewer 3 Report

Comments and Suggestions for Authors

This manuscript demonstrated the prescribing trend of PPIs and H2RAs in the UK, and evaluated the influence of COVID-19. The point of view is interesting. There are several  issues that should be addressed prior to the publication in “Healthcare”.

1. The problem of this study is that hospital prescribing data are excluded as described in discussion. The reason why hospital prescribing data are excluded and the ratio of GP prescribing to hospital prescribing should be described in discussion if possible.

2. It sems to be hard to understand for readers the differences between the contents of Table 3 and those of Table 4. I think it is better to make one Figure to show those differences.

3. The authors described the need for improved evidence dissemination and adherence to prescribing guidelines. How do you think the measures to disseminate the guidelines in clinical setting? These points should be described in discussion.

Round 2

Reviewer 3 Report

Comments and Suggestions for Authors

This revised manuscript has been extensively revised by considering the comments we previously raised. Now this manuscript appears acceptable for publication without changes any more in “Healthcare”.

Author Response

Dear Reviewer 3,

I sincerely appreciate your kind words and your recommendation for publication. Thank you for your thoughtful review.